# Developing a Mechanistic Approach to Sudden Death Prevention in Mitral Valve Prolapse

**DOI:** 10.3390/jcm11051285

**Published:** 2022-02-26

**Authors:** Brian P. Kelley, Abdul Mateen Chaudry, Faisal F. Syed

**Affiliations:** 1Division of Cardiology, University of North Carolina, Chapel Hill, NC 27599, USA; brian.kelley@unchealth.unc.edu; 2Department of Medicine, Southeast Health Medical Center, Dothan, AL 36301, USA; amchaudry@southeasthealth.org

**Keywords:** mitral valve prolapse, ventricular arrhythmia, sudden cardiac death, mitral regurgitation

## Abstract

Sudden cardiac death (SCD) from ventricular fibrillation (VF) can occur in mitral valve prolapse (MVP) in the absence of other comorbidities including mitral regurgitation, heart failure or coronary disease. Although only a small proportion with MVP are at risk, it can affect young, otherwise healthy adults, most commonly premenopausal women, often as the first presentation of MVP. In this review, we discuss arrhythmic mechanisms in MVP and mechanistic approaches for sudden death risk assessment and prevention. We define arrhythmogenic or arrhythmic MVP (AMVP) as MVP associated with complex and frequent ventricular ectopy, and malignant MVP (MMVP) as MVP with high risk of SCD. Factors predisposing to AMVP are myxomatous, bileaflet MVP and mitral annular disjunction (MAD). Data from autopsy, cardiac imaging and electrophysiological studies suggest that ectopy in AMVP is due to inflammation, fibrosis and scarring within the left ventricular (LV) base, LV papillary muscles and Purkinje tissue. Postulated mechanisms include repetitive injury to these regions from systolic papillary muscle stretch and abrupt mitral annular dysmotility (excursion and curling) and diastolic endocardial interaction of redundant mitral leaflets and chordae. Whereas AMVP is seen relatively commonly (up to 30%) in those with MVP, MVP-related SCD is rare (2–4%). However, the proportion at risk (i.e., with MMVP) is unknown. The clustering of cardiac morphological and electrophysiological characteristics similar to AMVP in otherwise idiopathic SCD suggests that MMVP arises when specific arrhythmia modulators allow for VF initiation and perpetuation through action potential prolongation, repolarization heterogeneity and Purkinje triggering. Adequately powered prospective studies are needed to assess strategies for identifying MMVP and the primary prevention of SCD, including ICD implantation, sympathetic modulation and early surgical mitral valve repair. Given the low event rate, a collaborative multicenter approach is essential.

## 1. Introduction

Mitral valve prolapse (MVP) refers to the systolic displacement of the mitral valve leaflet(s) of at least 2 mm above the annular plane into the left atrium [1]. Dr. John Barlow first described it in 1966 as an “electrocardiographic-auscultatory” syndrome before Dr. John Criley coined the term “mitral valve prolapse” later that same year [2]. Based on the features of myxomatous changes affecting the valve apparatus, MVP can be further classified into the following phenotypes: myxomatous or Barlow disease (or syndrome), fibroelastic deficiency (FED) and an intermediate state termed forme fruste [1]. It is a common valvulopathy with a prevalence in the general population of approximately 0.6–2.4%, and it represents the most common cause of mitral regurgitation (MR) in the Western world [3]. There is a wide spectrum of heterogeneity regarding its clinical presentation, ranging from asymptomatic to causing dyspnea, arrhythmia, palpitations, syncope and even sudden cardiac death (SCD) [4]. In a systematic literature review by Han et al. of 161 published cases of SCD in MVP patients, 55% had palpitations and 35% of patients had experienced a prior syncopal event [5]. The association with SCD has become of particular interest over recent years due to the ongoing challenges in the identification and management of this high-risk subset of MVP patients concealed within a large population of low-risk patients. The recognition of high-risk features using diagnostic modalities such as electrocardiography, echocardiography and magnetic resonance imaging have improved our understanding, but specific guidelines on the risk stratification and management of these MVP patients remain uncertain.

In his early description of the condition, Dr. Barlow stated, “The prognosis of this syndrome is uncertain, and sudden death may occur” [6]. Although MVP has a favorable prognosis overall, it has become well established that a cohort of at-risk patients also exists. These patients are susceptible to ventricular ectopy, which is now commonly labelled as arrhythmic mitral valve prolapse (AMVP). Among this subset of AMVP patients, there are some individuals at an especially high risk of arrhythmogenicity and who have a high potential for developing SCD, the malignant phenotype of MVP, or MMVP (see Figure 1) [7].

The proportion of MVP patients who have AVMP or MMVP and their proportional risk of developing SCD remains to be established. Published reports allow for an estimation of the annual risk of MVP-related SCD between 0.2% and 1.9% [4]. Kligfield et al. showed that the risk of SCD in patients with MVP without MR was 1.9 per 10,000 patients per year [8]. Nishimura et al. reported a higher incidence of SCD in those with redundant mitral valve leaflets (approximately 1.6% per year) compared to those without redundant leaflets (approximately 0.1% per year) [9]. Autopsy-based studies over the years have reported wide incidence rates for MVP-related SCD, in part due to population differences as well as the uncertain cause–effect relationship between the finding of MVP at autopsy and the arrhythmic SCD (see Table 1) [4,7,10,11]. In a systematic review by Basso et al., the proportional representation of MVP in SCD autopsy series ranged from 0 to 6.6% [12]. In the Oregon Sudden Unexpected Death Study, MVP was present in 2.3% of sudden cardiac arrest patients [9]. In the San Francisco POST SCD study, MVP was identified in 2% of SCD and 4% of sudden arrhythmic death [13]. 

Comparative studies have identified certain characteristics common to MVP patients with SCD or significant ventricular arrhythmia. These include complex and frequent ventricular arrhythmia on ECG or ambulatory monitoring, ECG T-wave inversion, bileaflet prolapse, mitral annular disjunction or dehiscence (MAD), rapid systolic movement of the lateral mitral annulus (Pickelhaube sign) and myocardial scarring on CMR [5,7,10,16,17]. In this review, we provide a comprehensive analysis of the data supporting, the mechanisms underlying and the potential approaches to mitigating SCD in MMVP. 

## 2. Patient Characteristics

MVP has been shown to have sex-differences in its presentation and outcomes, with women being more likely to have bileaflet prolapse with a greater degree of valve thickening and less likely to undergo valve surgery [18]. The latter is thought to be related to the left ventricular size—an important indicator for surgery—being underestimated in women. MVP has a female preponderance, with a M:F ratio of 1:1.2, and premenopausal women have classically been considered at higher risk for arrhythmias [7,18,19]. The significance of sex as a risk factor for MVP-related SCD remains controversial. Women comprise ~65% of individuals in published series of SCD from MVP [7]. However, some studies have not shown the female sex to be a risk factor for MVP-related SCD [20]. A larger cohort has reported that AMVP and MMVP can affect all age groups and both sexes [7,21].

Associations between the genetic substrate and MVP have been considered. Delling et al. described familial clustering with the prevalence of a family history of MVP in a large cohort of patients to be 20.4% and suggested that MVP has an autosomal dominant inheritance with variable expression [22]. A genetic linkage to MVP-related SCD was proposed by Narayanan et al. due to the younger age and fewer cardiovascular risk factors present in those with MVP versus those without MVP who sustained SCD [11]. Genetic variants affecting genes associated with arrhythmic and dilated cardiomyopathies (LMNA, SCN5A, RYR2, TTN and FLNC) have been considered as potential culprits in the arrhythmogenesis of bileaflet MVP-related SCD [23,24].

## 3. Mechanisms of Arrhythmogenesis

Events leading up to arrhythmic SCD are frequently described as an interplay of the following three factors: substrate, trigger and transient modulator (transient initiating event) [4]. In MVP, the SCD is mostly attributed to ventricular fibrillation (VF), with the proposed mechanism involving the excessive leaflet mobility that causes the mechanical stretch of the valve apparatus with subsequent stretch-induced fibrosis and ectopy [25,26]. The inferolateral base of the LV and papillary muscles are the most susceptible to the mechanical stretch forces exerted by the billowing leaflets [10]. Myocardial scars provide an arrhythmogenic substrate, while ventricular ectopy elicited from injured myocardium and mechanical stretch can act as potential triggers (see Figure 2) [27]. 

### 3.1. Ventricular Ectopy

Premature ventricular complexes (PVCs) and non-sustained ventricular tachycardia (NSVT) have been described in 43% of patients with MVP and may trigger VF, yet only about 6% of MVP patients develop SCD [9,20]. General population studies have demonstrated that PVC frequency is associated with increased all-cause mortality and increased risk for SCD [28,29]. Specifically in MVP, PVCs and VT arise from the vicinity of the mitral annulus and subvalvular apparatus, suggesting a causative association [26,30,31,32]. Some individuals, presumably those with more extensive myocardial scarring, may develop monomorphic VT [4]. It is possible that AMVP may be seen in conjunction with other pro-arrhythmic disorders that predispose to SCD, such as idiopathic VF. In cases of idiopathic VF, a Purkinje origin of the PVC and short-coupled PVCs are more likely to trigger VF [33,34]. However, findings from catheter ablation studies of VF-triggering PVCs arising from diseased Purkinje tissue on and in the vicinity of the LV papillary muscles point to structural mechanisms related to the MVP [26]. In addition, PVCs in MVP more frequently arise from the LV papillary muscles and fascicles than in non-MVP patients with frequent PVCs [30]. Sriram et al. reported a pattern of PVCs with alternating axes originating from the left ventricular outflow tract and papillary muscle/fascicular system in seven of nine MVP patients who experienced idiopathic out-of-hospital cardiac arrest [7]. In order to better risk stratify AMVP patients for SCD, an appreciation of how PVCs translate to VF is paramount. Our current understanding of mechanisms highlights the importance of PVC origin, frequency and coupling intervals, as well as the influence of associated structural and electrophysiological changes related to both PVC foci and remotely in the ventricle.

In addition to PVCs that originate from the papillary muscles, PVCs arising from the Purkinje tissue have also been implicated as a trigger for VF [35]. Several mechanistic studies have demonstrated the importance of Purkinje tissue in triggering and perpetuating VF [36]. Purkinje tissue forms a lattice network of cells that drape the endocardium and are insulated from the underlying myocardium until their peripheral arborization into muscle [37]. Physiologically, based on studies done on a rabbit and a canine cardiac tissue, Purkinje cells differ from working cardiomyocytes due to the presence of earlier repolarization, a more negative plateau potential and longer action potential duration and refractoriness [38,39]. Ionic currents and ion channel expressions also differ in Purkinje cells vs. ventricle myocytes (three vs. one calcium handling channels in endoplasmic reticulum) [40]. Therefore, proarrhythmic calcium-mediated mechanisms can lead to delayed afterdepolarizations of the Purkinje cells [41]. Stretch-induced ionic channel activation and ionic remodeling is also involved in causing Purkinje afterdepolarizations in heart failure and myocardial ischemia patients [42,43,44]. Arrhythmogenesis in papillary muscle (PM) can occur by both abnormal automaticity and re-entry circuit due to the unloading of depolarizing current from embedded Purkinje tissue [45]. 

A critical role for the Purkinje-papillary muscle complex has been shown through catheter ablation studies of VF triggers in MVP. In one study, Purkinje origin of the VF trigger was present in all 6 patients undergoing catheter ablation, of which 4 were from Purkinje tissue located on papillary muscles [26]. Furthermore, there was frequent electrogram evidence of both fascicular and myocardial disease in and around VA foci. Purkinje tissue is abundant on and around PM [46]. Such Purkinje involvement may occur in parallel to papillary muscle injury from repetitive papillary stretch secondary to prolapse, or diastolic interaction of mitral leaflets or cords with surrounding endocardium. Papillary muscle fibrosis has been reported in association with increased ventricular ectopy in AMVP, and PM ectopy can trigger VF [26,32,47]. The frequent finding of endocardial friction lesions located beneath the MV posterior leaflet at autopsy in patients with AMVP and SCD (present in 12/14, 86%) suggests that diastolic endocardial interaction with redundant mitral apparatus is also contributory [15].

### 3.2. Myocardial Abnormalities

Myocardial tissue abnormalities can result in arrhythmia from triggering and re-entry [48,49]. The causes of such myocardial abnormalities include mechanical stretch, fibrosis and remodeling. Myocardial stretch secondary to mitral valve prolapse and the abrupt tugging of papillary muscles and accompanying adjacent left ventricular musculature has been suggested to be arrhythmogenic in the absence of any tissue fibrosis [17,26]. Acute myocardial stretch can cause the shortening of action potential duration, a decrease in resting diastolic potential, and the development of stretch-activated early afterdepolarizations, all leading to abnormal automaticity [4,50]. Wilde et al. describe a case of a 56-year-old man with MVP and a 20-year history of life-threatening arrhythmia [27]. He was selected for valve replacement due to hemodynamic reasons. The electrophysiologic mechanism of his arrhythmia was studied preoperatively by using flunarizine, intraoperative epicardial and endocardial mapping, and, finally, the electrophysiologic characteristics of his excised PM were investigated ex vivo. The study suggests that the systolic stretch of fibrosed papillary muscles is involved in the etiology and origin of VE. Stretch applied during systole may lead to rapid repolarization or secondary depolarization, depending on stretch-activated ionic channels’ current-voltage characteristics. Stretch applied when resting potentials were positive to −30 mV causes transient repolarization, whereas stretch applied to potentials negative to −30 mV causes transient depolarization; the latter stretch-induced changes led to the development of arrhythmias. Findings on isolated myocardial tissue support this observation. Stretch applied in the plateau phase of AP leads to repolarization, whereas stretch applied during late systole results in afterdepolarization. Stretch-induced activation of the ATP-sensitive potassium channels leads to a shortening of AP and refractory period, which promotes re-entrant arrhythmias. Delayed afterdepolarizations (DADs) are less likely to be involved due to a lack of influence shown by flunarizine (a drug that selectively abolishes DADs). Electrophysiologic findings of PMs were further analyzed, which showed that conduction delays and stretch on such muscles exacerbates conduction delay, setting a stage for re-entry. Moreover, this study suggests that VT was caused by DADs, as indicated by its response to flunarizine and its inducibility with exercise [27]. 

Myocardial inflammation and fibrosis affecting the left ventricle has been demonstrated in AMVP by both autopsy studies as well as clinical imaging, both within the papillary muscles as well as at the basal posterolateral segment of the left ventricle [51,52,53]. Several mechanisms leading to chronic repetitive mechanical injury associated with the mitral valvular abnormality have been postulated. Superficial plaque lesions are present beneath the MV posterior leaflet at autopsy in patients with AMVP and SCD (present in 12/14, 86%), a location consistent with repetitive diastolic interaction from redundant subvalvular apparati, whilst repeated systolic papillary muscle traction and excessive motion of the posterobasal segment associated with systolic annular excursion are postulated to cause the fibrosis in these respective areas [15]. The resulting inhomogeneous anisotropy is a critical substrate for reentry, whilst electrical decoupling may predispose to trigger activity and promote the dispersion of refractoriness. 

Myocardial remodeling can be seen as secondary to significant MR as well as a PVC-induced cardiomyopathy [54]. An increase in LV mass as well as a larger mitral annular dimension have been reported on autopsy studies of MVP patients suffering from SCD, although it remains unclarified whether this was secondary to the aforementioned changes or an MVP associated cardiomyopathy [5,55]. 

### 3.3. Autonomic Nervous System

Autonomic dysfunction may contribute to arrhythmogenesis in MVP patients. A sympathetic tone plays an important role in both triggering arrhythmias and initiating scar-related reentrant arrhythmia and has been proposed as playing a role in MVP-associated ventricular arrhythmias [56]. Among the 10 patients with AMVP who survived cardiac arrest, Sriram et al. reported 2 of them (20%) as having acute emotional or physical stress at the time of the event [7].

## 4. Electrocardiography (ECG) and Holter Monitor

Rhythm disturbances frequently seen in AMVP patients include repolarization abnormalities and complex ventricular ectopy. Inferolateral T-wave inversions (TWI) are the most frequent repolarization abnormality observed [7,18]. More than 75% of MVP-related SCD patients have biphasic T-waves or TWI in the inferior ECG leads [10]. However, this is also seen in up to 40% of all MVP patients. The mechanism underlying ECG repolarization abnormalities seen in AMVP remains unclarified, with postulated mechanisms including T-wave memory associated with frequent PVCs, LV remodeling or LV fibrosis. Evidence of scarring in the basal inferolateral segment of the LV on unipolar voltage mapping was associated with TWI in a recent study [57]. Figure 3 shows the case of a patient with bileaflet MVP in which TWI and PVCs are present on pre-PVC ablation ECG (A) with subsequent improvement on post-ablation ECG (B). Patients with AMVP have increased LV mass, which is associated with repolarization abnormalities including prolonged QT. Although some studies report as many as 75% of MVP patients as having QT prolongation, others have found no association between MVP and QT prolongation [8,56]. Patients with MVP and concomitant congenital long-QT syndrome (LQTS) are considered to be at an especially high risk of ventricular arrhythmia. In one single-center series of 754 LQTS patients, 5 had associated AMVP [58]. Breakthrough VA and ICD shocks were much more frequently seen in these patients as compared to the rest of the LQTS cohort. An increase in QT dispersion, independent of QT prolongation, has been reported to be associated with increased VA risk in MVP. Kulan et al. performed a study with 64 MVP patients and 80 control subjects, which showed a strong correlation between QT dispersion and complex PVCs [59].

Abnormalities of repolarization may also account for the increased incidence of AMVP associated SCD in women. It is well known that torsades de pointes (TdP) is more frequently seen in women [60,61,62]. In a modeling study, Yang et al. presented hormone effects on ventricular cell and tissue dynamics comprised of the Faber-Rudy computational model, which predicted changes in action potential duration (APD) at different stages of the menstrual cycle, consistent with clinically observed QT interval fluctuations, with hormonally-induced APD prolongation allowing for the initiation of reentry and TdP [60].

Complex ventricular ectopy in AMVP generally refers to the presence of one or more of the following: ventricular bigeminy, PVC couplets, nonsustained ventricular tachycardia (NSVT) and sustained VT. The dominant PVC morphologies are those arising from the papillary muscle region or outflow tract (see Figure 4C,D for case example) [7]. The ectopy is presumed to be at least in part due to regional stretch, either because of direct mechanical forces or triggered activity due to damaged tissue with abnormal calcium handling and evoked DADs. PVC couplets with alternating axes are seen frequently in MVP SCD survivors (MMVP) [7]. Their prevalence in the general MVP population has not been systematically investigated. Whether PVC coupling intervals are predictive is unknown, but short-coupled PVCs have been reported in other forms of PVC-induced ventricular fibrillation [63].

## 5. Echocardiography

The echocardiogram serves as a valuable tool for assessing for abnormalities involving the valvular apparatus and myocardium. These include mitral leaflet morphology, mitral regurgitation (MR), mitral annular disjunction or dehiscence (MAD), the Pickelhaube sign and global longitudinal strain (GLS). In particular, the echocardiographic presence of MAD and the Pickelhaube sign have become popular in the risk assessment, owing to their ability to serve as predictors of scars.

Bileaflet morphology has frequently been identified as a high-risk feature of SCD. Among a cohort of 24 patients with an implantable cardioverter-defibrillator (ICD) who experienced idiopathic out-of-hospital cardiac arrest without an identified etiology after comprehensive workup, 10 (42%) were found to have bileaflet MVP [7]. However, a subsequent study showed that isolated bileaflet MVP did not appear to significantly increase the risk of SCD compared to single-leaflet MVP [64].

MR is often seen in conjunction with MVP and represents an important consequence of the disorder. In a cross-sectional population-based study by Freed et al., 70% of MVP patients were found to have some degree of MR [65]. However, severe MR was uncommon, identified in only 4% of MVP patients. Patients with both MR and MVP have been shown to have more ventricular arrhythmias than those with MVP alone [8]. Among patients with MVP, there appears to be an increased risk of SCD when the MR is secondary to flail leaflet [66]. Although MR severity has been associated with VAs in MVP, there is conflicting evidence. Due to it being a strong risk factor for SCD in and of itself, MR is a confounder and its presence is therefore challenging to interpret. Turker et al. reported moderate to severe MR to be an independent predictor of VA [67]. Thus, the increased risk of SCD associated with MR may be directly related to the valvular regurgitation, rather than MVP itself. Additionally, MVP-related SCD is known to occur in patients with minimal to mild MR [10,67]. Essayegh et al. found no association between severe VA and MR severity or LV ejection fraction [20]. 

MAD is defined as a ≥1 mm distance between the left atrial wall-mitral leaflet junction and the top of the LV wall during end-systole that can be observed on both echocardiography as well as CMR [68]. It represents the hypermobility of the mitral valve apparatus in which there is a paradoxical systolic increase of the mitral valve annulus. MAD has been reported to be present in about 30% of MVP patients and associated with AMVP in several studies, and it has been described as a possible mechanism leading to fibrosis [69]. It has been shown to be significantly longer in those with LGE on MRI compared to those who did not have LGE on MRI. Dejgaard et al. showed that MAD was associated with VAs even in absence of detectable MVP, thus suggesting that MAD itself is an arrhythmogenic entity [68]. Essayagh et al. reported MAD and redundant leaflets to be the highest predictors of arrhythmia in MVP, with OR of 6.97 (95% CI: 3.31–14.78, *p* < 0.0001) and 3.85 (95% CI: 1.90–7.85, *p* < 0.0001), respectively [20]. Despite this association with AMVP, it is worth noting that a recent study by Toh et al. has shown MAD to be a common finding in patients without mitral valve prolapse, present in 94 of 98 cases (96%) of subjects with normal mitral valves assessed using CT imaging [70].

Tissue Doppler imaging (TDI) to assess for the Pickelhaube sign, which suggests that there is a high velocity on the mitral valve annulus, is also useful in the evaluation of AMVP. The tugging of the PM papillary muscle in mid-systole by myxomatous prolapsing leaflets causes the adjacent posterobasal LV wall to be pulled sharply towards the apex, leading to a high-velocity systolic signal on TDI that resembles a Pickelhaube. In a 21-patient sample by Muthukumar et al., two groups of BiMVP myxomatous MVP were compared—one with a positive Pickelhaube sign and the other without a Pickelhaube sign [17]. Those with Pickelhaube sign were more likely to have VA (67% vs. 22%) and LGE on CMR (33% vs. 0%). When a systolic peak velocity of ≥16 cm/s was used, patients were more likely to have fibrosis and complex VAs. 

Speckle-tracking echocardiography (STE) can be used to quantify global longitudinal strain (GLS). Studies have shown that patients with MVP have reduced radial and circumferential strain values in the basal and mid-ventricular inferolateral segments relative to other segments. Whereas GLS is a measure of the overall longitudinal contraction of the myocardium, STE-derived mechanical dispersion is a heterogeneous ventricular contraction parameter. Mechanical dispersion is predictive of arrhythmic risk in several conditions, including ischemic heart disease, hypertrophic cardiomyopathy, dilated cardiomyopathy and arrhythmogenic right ventricular cardiomyopathy. Ermakov et al. differentiated 59 MVP patients into arrhythmic and non-arrhythmic MVP patients [71]. The study showed higher mechanical dispersion in patients with arrhythmic MVP and even higher dispersion in patients with secondary ICD implantation.

## 6. Cardiac Magnetic Resonance (CMR) Imaging

Cardiac magnetic resonance (CMR) has gained popularity as a noninvasive imaging modality for the detection of high-risk features in MVP patients that can be used in risk stratification [10,72]. These features include MAD, systolic curling and fibrosis (myocardial and/or papillary scar) manifested as late gadolinium enhancement (LGE). Figure 4A,B show an example of a patient with AMVP who had evidence of MAD on CMR. Fibrosis is most often detected in the papillary muscles (predominantly the posteromedial) or inferobasal region of the LV under the posterior mitral valve leaflet [6]. Constant Dit Beaufils et al. observed replacement myocardial fibrosis in 110 of 400 (28%) MVP patients who underwent CMR (91 with myocardial wall including 71 with basal inferolateral wall; 29 with papillary muscle) [73]. Additional fibrosis patterns include subendocardial fibrosis and diffuse interstitial fibrosis (T1 mapping/ECV). Given the association of myocardial fibrosis with SCD, the detection of endocardial fibrosis is an essential step to stratify high-risk SCD patients. A study with 41 participants showed reduced post-contrast times in patients with MVP, suggesting diffuse LV myocardial fibrosis. Kitkungvan et al. studied 356 patients who underwent CMR, dividing them into MVP and non-MVP patients, and he found a significantly higher prevalence of LV fibrosis in the MVP patients group [74]. However, another recent study failed to find any significant associations between complex VA in MVP patients and T1/ECV values [75]. Scatteia et al. evaluated the cardiac MRI of 47 consecutive patients with MVP compared with a group of patients without MVP and noted significantly lower signal intensity in papillary muscles in the MVP group [51].^.^

Systolic curling refers to the unusual systolic motion of the posterior mitral ring on the adjacent LV myocardium and correlates well with scars on CMR [10,16,72]. Systolic movement of the posterior mitral annulus, primarily downwards with little, if any, anterior motion, results in a curled appearance when visualized in real-time motion. The mechanism of systolic curling is not fully established but may be related to distracting forces on the LV lateral wall from the myocardial insertion of a prolapsing papillary muscle, similar to the mechanism thought to underlie the Pickelhaube sign.

A recent CMR study of 92 MVP patients demonstrated that apical papillary muscle insertion, seen in 53 of the 92 patients, was more frequently associated with MAD, abnormal LV strain, increased lateral wall remodeling and more frequent papillary fibrosis, lending further support to the idea that AMVP may be associated with a unique morphological variant of MVP [76].

## 7. Risk Stratification

Although there have been advancements in risk stratifying the MVP patient, it remains a work in progress due to the elusiveness of identifying the individual at risk of SCD. In those with complex VAs on electrocardiograms and/or evidence of abnormal myocardial stretch on echocardiograms, further evaluation with CMR can be considered. Additional approaches—albeit with limited supporting data—have included extended ECG monitoring (such as with a 24-h Holter monitor, a 7-day external event recorder or an ILR) for patients with unexplained syncope and an electrophysiology study (EPS) to assess for inducible sustained ventricular arrhythmia. 

Some, including Basso et al. and Miller et al., have proposed algorithms for risk stratification [4,12]. When multiple high-risk features are present, EPS with ventricular extra-stimulus testing can be performed to help further risk stratify patients. However, there are no established guidelines for the interpretation of the results. Proposed criteria for a positive study by Miller et al. include the following: sustained monomorphic VT induced with up to 3 ventricular extrastimuli, or polymorphic VT or VF induced with up to 2 ventricular extrastimuli [4]. If the EPS study is positive, then consideration of an ICD is warranted. If the EPS is negative, it has been suggested to pursue long-term rhythm monitoring with a device such as an implantable loop recorder (ILR). 

However, despite these recommendations, there is limited evidence for the use of EPS and programmed electrical ventricular stimulation (PVS) to risk stratify patients with MVP. Rosenthal et al. performed an EP study with PVS on 20 MVP patients compared to 12 patients without structural heart disease as controls [77]. Nine of the twenty MVP patients had a positive response to ventricular stimulation, whereas only 1/12 control patients responded to ventricular stimulation. Morady et al. performed programmed ventricular stimulation analysis on 36 patients with MVP [78]. Twenty of these patients had cerebral symptoms (i.e., syncope or pre-syncope) or documented NSVT or PVCs, and they noted that 65% of these patients had inducible VT/VF. In another study by Vergara et al., among 23 patients, including 5 with syncope and 4 with a history of SCD, only 4 patients had sustained VA on EPS with programmed ventricular stimulation (PES), including 1 of the 4 who had a history of SCD [57]. Whilst EPS may have value in identifying high-risk patients (i.e., those with ventricular scars associated with inducible sustained VA), this study suggests that the induction of sustained VA on EPS with PES is an insensitive marker.

Several studies have used Holter monitors to assess PVC burden and risk stratify patients. Essayegh et al. described NSVT >180 bpm as strongly predictive for the combined endpoint of VT/VF, ICD implantation and mortality compared to patients with no ventricular arrhythmia on ambulatory monitoring [20]. An ILR can be utilized if syncope is present in MVP patients to assess the rhythm long-term and its correlation with symptoms. The detection of frequent or complex PVCs alone is not enough to deem a patient as high risk for SCD, but it does suggest the need for further evaluation and risk stratification, whilst the finding of VF or sustained VT should prompt the consideration of ICD implantation for sudden death prevention.

We propose a risk-stratification algorithm that entails a multimodal approach to distinguishing low-risk versus high-risk patients (see Figure 5). While reassurance can be provided to the low-risk patient, ICD placement should be strongly considered for the high-risk patient. The diagnostic dilemma will be stratifying the patient whose risk lies somewhere in between. Based on currently available data, these patients should be considered to have an undefined risk and encouraged to have a routine clinical follow-up. 

## 8. Management

The approach to the management of MVP is an ongoing challenge owing to the lack of standardized risk-stratification models and guidelines. Individuals warranting protection against sudden arrhythmic death should have ICD implantation. We recommend ICD implantation for patients surviving cardiac arrest or for those who have sustained scar-related VT or syncope with documented polymorphic VT. The utility of anti-arrhythmic medical therapy, ventricular arrhythmia ablation, mitral valve repair and sympathetic modulation should be individualized to treat symptomatic ventricular arrhythmia, recurrent ICD shocks or significant mitral valve dysfunction. An individualized approach based on patient-physician preference and an assessment of the risk-benefit profile is recommended for those with symptomatic PVCs or PVC-induced cardiomyopathy and for patients who are asymptomatic but have a high PVC burden. It is important to note that these interventions are unlikely to obviate the risk of SCD. While anti-arrhythmic medications may suppress VA and reduce ICD shocks, they should not be considered protective against SCD. Results from the SCD-HeFT trial showed no benefit from amiodarone on mortality in at-risk cardiomyopathy patients [79]. Similarly, catheter ablation may successfully abolish VA, including VF triggering PVCs, but the available data suggest a high rate of late recurrence, presumably from ongoing repetitive injury to the mitral annulus, papillary muscles and Purkinje tissue from MVP. Theoretically, mitral valve repair may relieve, leading to a reduction in VAs. However, data supporting this is limited to small retrospective case series and isolated case reports. Multicenter randomized controlled trials are needed to assess recent advancements in catheter ablation technology as well as to address the potential use of early mitral valve surgery in selected patients. These therapies will be further discussed below. 

## 9. Noninvasive/Medical Therapy

For low-risk MVP patients, conservative management with surveillance cardiac monitoring with or without medical therapy is recommended. Medications routinely used for ventricular arrhythmias, such as beta-blockers, calcium channel blockers and other anti-arrhythmics, may be trialed [80]. Preventative interventions are not well understood, but the avoidance of stimulants (e.g., caffeine, tobacco, alcohol) that may serve as potential transient modulators is generally encouraged. Evidence regarding recommendations for other situations associated with elevated catecholamines, such as high-intensity exercise and the perioperative period, is limited. 

## 10. Implantable Cardioverter-Defibrillators (ICDs)

ICD implantation is indicated in those surviving SCD. Bumgarner et al. studied 43 patients with isolated phenotypic MVP and significant PVCs, and 13 (30%) underwent secondary ICD placement after VF [31]. Similarly, Syed et al. studied 14 phenotypic arrhythmic MVP patients, and 6/14 (42%) had secondary ICD placement for cardiac arrest [26]. Although generally recommended for secondary prevention, there is currently no sufficient data to guide the utility of ICD implantation for primary prevention in those with high-risk features. 

The indication for ICD placement as a means of primary prevention is much less clear. As discussed previously, some have recommended that AMVP patients undergo an EPS for further risk stratification [4,12]. In those with spontaneous or inducible sustained ventricular arrhythmia, it is prudent to take the clinical context into account, especially as the patient may have incidental idiopathic VT, such as outflow tract, papillary muscle or fascicular VA. Therefore, we argue for reserving an EPS for those with ventricular scars and inducible monomorphic VT. Yokokawa et al. depicted the value of inducible VT in individuals with structural heart disease who underwent PVC ablation [81]. Extrapolating their results to MVP patients, if a patient has evidence of myocardial scars on either CMR or electro-anatomical mapping, as well as induced sustained VT from an area of the scar, even if arising from the outflow tracts, then ICD implantation is reasonable. Due to prior studies being underpowered, a collaborative effort with multicenter prospective studies is needed to provide standardized risk stratification.

## 11. Ventricular Ablation

Due to the multifactorial nature of PVCs in AMVP, catheter ablation has mostly been reserved for patients with malignant PVCs and/or in cases with an electrical trigger of VT/VF that can be mapped (see Table 2). A Purkinje-mediated PVC was identified as the VF trigger by Syed et al. in six bileaflet MVP patients with a prior history of cardiac arrest who underwent catheter ablation [26]. The ablation of clinically dominant ventricular ectopy foci was shown to improve symptoms and reduce appropriate ICD shocks. Arrhythmic bileaflet MVP in the study was characterized by fascicular and papillary muscle PVCs that trigger VF, suggesting a central role of the Purkinje fiber system in the condition. However, there was recurrence with 6 of the 14 patients needing repeat ablation. A more recent study by Marano et al. included all MVP subtypes and had a longer follow-up period (median of 9 years) [82]. Although acute procedural success was demonstrated, there was a subsequent recurrence of VAs as well. This is likely related to the multifocality and progression of the complex ventricular ectopy. Due to this risk of recurrence, ablation should not be used as a means to simply prevent SCD but offers an important therapeutic modality for reducing ICD discharges, symptomatic PVCs and frequent PVCs in the setting of cardiomyopathy based on current guidelines [83]. However, the exact role of catheter ablation in the management of MVP-related SCD remains uncertain at this time.

El-Eshmawi et al. described empiric surgical cryoablation of the papillary muscles in three MMVP patients with frequent PVCs undergoing mitral valve surgery [85]. They observed a median reduction of 97.9% in the PVC load after PM cryoablation. As with catheter ablation, evidence supporting the use of adjunctive surgical cryoablation at the time of mitral surgery is insufficient, and its impact on SCD risk remains unclear. 

## 12. Mitral Valve Surgery

As discussed earlier, the most common cause of SCD in MVP patients is ventricular arrhythmias. Fatal ventricular arrhythmia (VF/VT) are propagated by PVCs originating from scar tissue in the left ventricle (the most common origin sites are papillary muscles and fascicles). This scar tissue is developed from excessive mechanical stretch due to the mitral valve’s prolapse in the left atrium. Theoretically, this fatal complication can be avoided by timely repair of the mitral valve before scar tissue develops. There have been reports of MVP patients with refractory Vas undergoing MV repair for severe MR and having a subsequent disappearance of the arrhythmia burden (see Table 3) [86]. Surgical repair has also been described as leading to a reduction in the number of appropriate ICD shocks, but it did not significantly affect the overall PVC burden [87]. In another study looking at 32 bileaflet MVP patients, ventricular ectopy burden was unchanged after mitral valve surgery, but there was a reduction observed in younger patients—possibly related to the more diffuse extent of fibrosis with age [26].

## 13. Areas of Future Research

The autonomic nervous system is known to play a central role in the genesis and maintenance of arrhythmias [89]. Neuromodulation, which is mostly designed to augment parasympathetic tone and suppress sympathetic tone, has emerged as an attractive treatment option for refractory VA. In a systematic review by Hawson et al., renal denervation was shown to significantly reduce ICD therapies and VA burden in patients with refractory VA or electrical storm [90]. Okeagu et al. recently described the successful use of sympathetic denervation in a patient with Lamin A/C (LMNA) cardiomyopathy who underwent thoracic bilateral stellate ganglionectomy [91]. However, the potential role for neuromodulation in MVP-related SCD has not yet been explored. 

The role of circulating biomarkers and genetic testing in risk stratification is still unknown. As mentioned previously, mechanical stress-induced fibrosis is believed to be play a central role in the pathophysiology. This may involve the upregulation of transforming growth factor-β1 (TGF-β1) and other growth factors. The soluble suppression of tumourigenicity-2 (sST2) and TGF-β1 has been studied as a potential marker of ventricular arrhythmia in patients with MAD. Circulating sST2 levels were higher in patients with MAD and ventricular arrhythmia compared with arrhythmia-free patients [92]. MicroRNAs (miRNAs) have recently been shown to have an association with MVP pathology, which may help identify patients and pathways involved in the pathophysiology of the disorder [93].

The association of fibrosis on imaging with the risk of sudden cardiac death continues to be investigated. In a recent study by Lee et al., the presence of LGE on CMR was found to be an independent predictive factor for SCA or VA in MVP patients [94]. However, it is worth noting that CMR carries a degree of operator dependence in addition to a dependence on the overall quality of the study for the detection of fibrosis. The utility of CMR in predicting the risk of sudden cardiac death remains unclear, and longitudinal studies are needed to better elucidate its prognostic value.

Miller et al. recently described the use of a hybrid PET-MRI system in a cohort of 20 patients with AMVP [95]. Focal or focal-on-diffuse uptake of fluorine 18-labelled (18F) fluorodeoxyglucose (FDG)—“PET positive”—was detected in 17 of 20 patients (85%). The FDG uptake coexisted with areas of LGE (PET/MRI positive) in 14 patients (70%). PET-FDG uptake, a surrogate for myocardial inflammation and/or ischemia, may represent arrhythmogenic substrate and demonstrate an association between AMVP and myocarditis. This suggests that the pattern of injury and myocardial fibrosis is linked by an inflammatory phase. It is uncertain whether this is persistent or cyclical and, as with other myocarditis conditions, if it is an additional proarrhythmic element. The role of systemic inflammation and environmental influences have not yet been studied in AMVP. Further investigation is still needed to determine the role and significance of hybrid PET-MRI in risk stratification and prognostication. 

There may be a role for artificial intelligence in better identifying patients with AMVP and MMVP when applied to existing investigational approaches. Recently, such an approach was able to predict the presence of complex ventricular ectopy on ambulatory monitoring from a resting 12-lead ECG with an area under the curve (AUC) of 0.81 [96]. The extension of these approaches to imaging and electrophysiological findings has the potential to predict SCD in MVP. 

Early surgical intervention may also serve a role by modifying the underlying electrophysiologic substrate and warrants further study. 

## 14. Conclusions

MVP is a common, heterogenous disorder comprised of a spectrum of many low-risk patients on one end but also a high-risk subset on the other end (MMVP). The incidence of SCD in patients with MVP is not clearly established. It is usually due to VF and associated with excessive leaflet mobility inducing mechanical stretch of the inferobasal wall and papillary muscles and diastolic interaction of the leaflets or chordae with the ventricular endocardium, leading to inflammation and scarring. Fibrosis in these areas along with MAD and systolic curling of the mitral leaflets are the proposed pathophysiologic mechanisms for SCD. Not only has the identification of those with MMVP, who are at a high risk of SCD, proven challenging, but the management of MMVP also remains uncertain. This is especially true for determining appropriate ICD placement as well the utility of early mitral valve repair and prophylactic catheter ablation. Based on our current understanding, patients with high-risk features such as bileaflet involvement, complex ventricular ectopy, MAD, systolic curling, ventricular fibrosis and increased ventricular mass should undergo further risk stratification in order to mitigate the risk of SCD. The implementation of a multimodality approach with multicenter RCTs will be paramount in achieving this. 

## Figures and Tables

**Figure 1 jcm-11-01285-f001:**
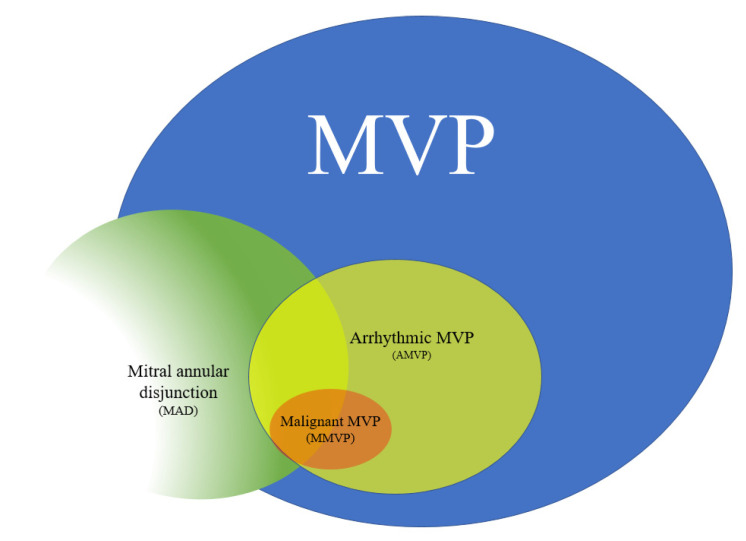
Mitral valve prolapse (MVP) represents a common valve disorder with a subgroup at elevated risk of arrhythmias, referred to as arrhythmic MVP (or AMVP). Within AMVP, there is a subset of patients at risk for developing SCD, termed malignant MVP (or MMVP). Mitral annular disjunction (MAD) has established itself as an important risk factor for AMVP and MMVP.

**Figure 2 jcm-11-01285-f002:**
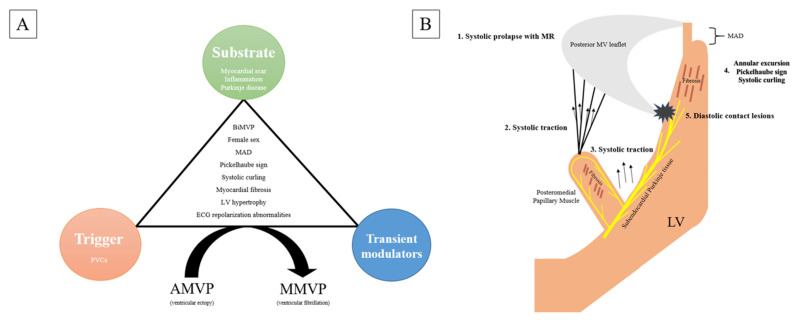
Mechanisms of arrhythmogenesis. (**A**) An interplay of substrate, trigger and transient modulators in the setting of other high-risk features involved in MVP-related SCD. (**B**) Prolapse leads to traction-induced injury of the mitral valve apparatus and its associated Purkinje network.

**Figure 3 jcm-11-01285-f003:**
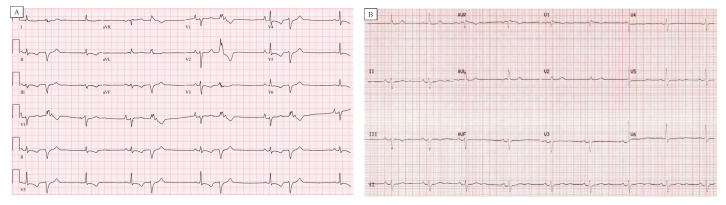
12-lead ECG in a patient with bileaflet MVP. (**A**) ECG reveals TWI in multiple leads as well as bigeminal PVCs that have a posteromedial papillary muscle morphology. (**B**) Repeat ECG following PVC ablation demonstrates an improvement in TWI.

**Figure 4 jcm-11-01285-f004:**
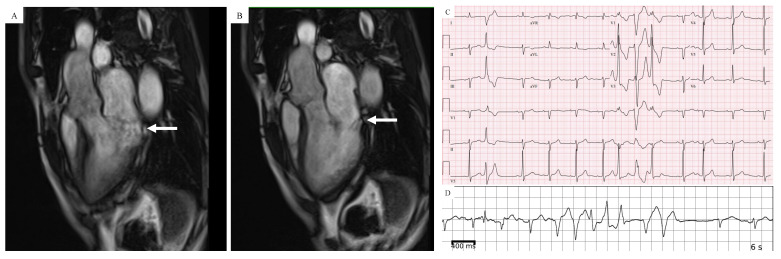
Patient with AMVP who has evidence of mitral annular disjunction (MAD) by CMR in a 3-chamber long-axis view in systolic frame (**A**) and diastolic frame (**B**). (**C**) 12-lead ECG showing PVCs with alternating axes. (**D**) Complex ventricular ectopy captured on ambulatory ECG monitoring.

**Figure 5 jcm-11-01285-f005:**
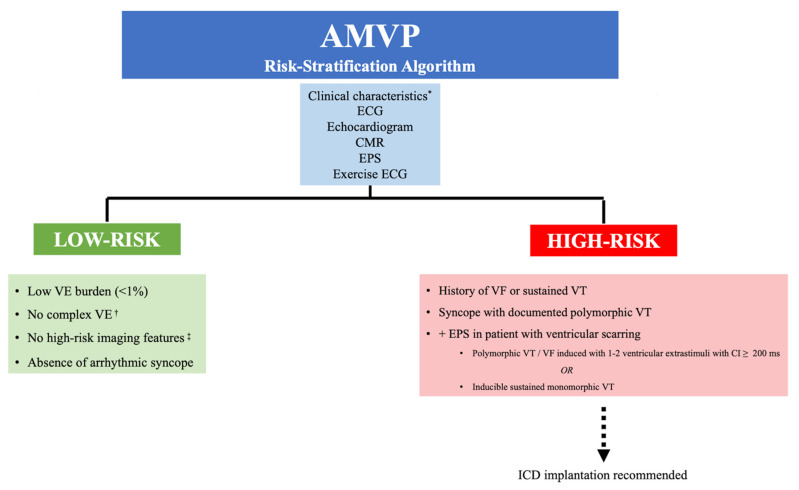
Proposed risk-stratification algorithm for AMVP patients. AMVP: arrhythmic mitral valve prolapse. CMR: cardiac magnetic resonance imaging. CI: coupling interval. ECG: electrocardiogram. EPS: electrophysiology study. ICD: implantable cardioverter-defibrillator. VE: ventricular ectopy. VF: ventricular fibrillation. VT: ventricular tachycardia. * Clinical characteristics such as female sex and arrhythmic syncope. ^†^ Complex VE is defined as having ≥1 of the following: ventricular bigeminy, PVC couplets, nonsustained ventricular tachycardia (NSVT) and sustained VT. ^‡^ High-risk imaging features include myxomatous bileaflet prolapse, mitral annular disjunction, abnormal annular motion (Pickelhaube sign, systolic curling, annular expansion), increased ventricular mass and CMR features of ventricular scarring.

**Table 1 jcm-11-01285-t001:** Autopsy studies of SCD in AMVP.

Study (Ref)	Number of Patients	Increased Cardiac Mass (%)	LV Dilatation (%)	LV Hypertrophy (%)	LV Fibrosis/Scar (%)	PM Fibrosis (%)
Dollar et al. [14]	56	100	5/48 (10)		5/48 (10)	-
Han HC et al. [5]	70	69	-	49/70 (70)	52/70 (74)	2/70 (2)
Chesler E et al. [15]	14	50	-		11/14 (78%)	-

**Table 2 jcm-11-01285-t002:** Outcomes of catheter ablation of ventricular arrythmia in AMVP.

Study Name	No. of Participants	Follow Up Time (Years)	Repeated Ablation (*n*)	Recurrent PVCs (*n*)	Recurrent VT/VF (*n*)	Successful Ablation (*n*)
Marano et al. [82]	15	9	-	3	5	7
Syed et al. [26]	8	~2	3	3	-	5
Lee et al. [84]	9	6	3	2	-	7
Bumgarner et al. [29]	30	2.5	-	-	11	20
Total	62		6/17 (35%)	8/32 (25%)	16/45 (35%)	39/62 (62.9%)

**Table 3 jcm-11-01285-t003:** Reduction in ventricular arrhythmia and ICD shocks after surgery in AMVP.

Study Name	No. of Patients (*n*)	ICD in Place	PVC Reduction (*n*, %)	VF/VT Reduction (Events Per Person-Year)	ICD Shock Reduction
Naksuk et al. [88]	32	No	>10% (*n* = 17, [53%])	-	-
Vaidya et al. [87]	8	Yes	-	~70% (0.6 to 0.14)/~80% (0.4 to 0.05)	~80% (0.95 to 0.19)
El-Eshmawi et al. [85]	3	Yes	97.7% (*n* = 3, [100%])	-	-

## Data Availability

Not applicable.

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
