# Peer review of "Developing a Mechanistic Approach to Sudden Death Prevention in Mitral Valve Prolapse"

_jcm, 2022, doi:10.3390/jcm11051285_

Round 1

Reviewer 1 Report

The mansucript is an interesting review of a field in current cardiology that warrants further interest. Although interesting some revisions are needed to appropriately adress the scientific data currently available.

Throughout the manuscript revision for correctness of English language and style is needed.

Figure 1 is misleading and unclear. It lookgs like malignant MVP only would exist if MAD is detected but of course there are other indicators as well? Please revise accordingly.

Paragraph 8 - Management needs revisions, specifically for the following points:

lines 412 to 415 - I would recommend to not make any conclusions different than current GL for the MVP group, data is not available for this and PVC trigger ablation may be helpful in this cohort as well.

line 421 - how are low-risk MPV patient defined? please clarify. 

line 430 to 431 - ICDs are not specifically indicated in pts. with inducible VA, actually depends on the clinical status. What about outflow tract VTs?

lines 454 to 458 - again ablation may be helpful in ablation of PVCs, even though this may not have an effect on risk stratification? Please clarify. Ablation should not be considered as palliative as - if done in experienced centers - may provide clinical benefit and reduce VAs.

lines 471 to 473 - is there any data on the protective value of MV surgery? Please clarify and indicate!

line 524 - future research also includes the preditive value of CMR in this cohort and what an effect will arise from LGE detection?

Please try to formulate a risk stratification algorithm to apply different steps for risk evaluation in MVP patients?!

Reviewer 2 Report

The author present a review on sudden cardiac death in patients with mitral valve prolapse.

The paper is well written and comprehensive. 

Although it is unfortunately true that further studies should be carried out as no specific recommendations have been proposed to help decisions in the context of MVP and SCD, it would be interesting if the authors could expand the paragraph on "risk stratification" adding a few summarizing lines on information on ecg changes and imaging alteration to look for (i.e picklelhaube sign, GLS alterations which were described in previous paragraphs).

There are minor spelling mistakes which should be corrected.
